# Whole-genome sequencing and phylogenetic analysis of *Salmonella* isolated from pullets through final raw product in the processing plant of a conventional broiler complex: a longitudinal study

Yagya Adhikari,[1] Matthew A. Bailey,[1] Steven Kitchens,[2] Pankaj Gaonkar,[2] Luis R. Munoz,[1] Stuart B. Price,[2] Dianna V. Bourassa,[1] Laura Huber,[2] Richard J. Buhr,[3] Kenneth S. Macklin[4]

**ABSTRACT** *Salmonella* are Gram-negative, rod-shaped, entero-invasive foodborne bacteria and are frequently detected in chicken houses and facilities of poultry broiler complexes. The objective of this study was to determine the prevalence, critical entry points, and movement pattern of *Salmonella* along different stages of a complex. A total of 1,071 environmental samples were collected from 38 production houses (8 pullet, 10 breeder, and 20 broiler), a hatchery, 6 transport trucks, and a processing plant. Samples were screened with 3M Molecular Detection System and were further processed for the confirmation of results. Whole-genome sequencing and phylogenetic analysis were performed to determine genetic relatedness among bacterial strains. Using multivariable model, the odds ratios and 95% confidence limits were compared for stages, sample types, environments, and seasons ($a < 0.05$). Altogether 18% of samples and 42% of production houses tested *Salmonella* positive. Interestingly, the odds of *Salmonella* detection were more likely ($P ≤ 0.001$) in facilities like hatchery, transport, and processing plant as compared to production farms such as pullet, breeder, and broiler farms. The predominant serotype identified was *S.* Kentucky followed by *S.* Enteritidis, *S.* Typhimurium, *S.* Johannesburg, *S.* Montevideo, *S.* Mbandaka, *S.* Newport, *S.* Senftenberg, *S.* Inverness, *S.* Ohio, *S.* Uganda, and N/A (9:z29:-). Phylogenetic analysis showed strong genetic relationship among bacterial strains isolated from different stages. It also suggests diverse movement patterns of bacterial strains and possibility of multiple critical points for bacterial pathogens entering the complex. From the above results, we can conclude that *Salmonella* from chicken houses/facilities' environment can enter the broiler complex and can potentially contaminate the final raw product in the processing plant. A multifaceted comprehensive control strategy focusing on both facilities and production farms might be essential for improved control strategies.

**IMPORTANCE** *Salmonella* continues to be the leading human bacterial foodborne pathogen, a serious food safety concern. The major challenges are to reduce the risk of introduction or spread of such bacteria in flocks, to minimize the persistence of such bacteria within the broiler complex, and to achieve USDA FSIS final product standards at the processing plants. Not well understood are the possible entry points and movement patterns of *Salmonella* along different stages of an integrated broiler complex. For this study, environmental sampling was considered from parent pullets through the final raw product at the processing plant, and SNP-based analysis of *Salmonella* isolates was conducted to determine the genetic relatedness and movement patterns. Interestingly, the samples from facilities (hatchery, transport, and processing plant) were more likely to be contaminated with *Salmonella* as compared to production farms (parent pullets,

**Peer Reviewers** Nurul Hawa Ahmad, Universiti Putra Malaysia, Selangor, Malaysia; Ben Davies Tall, FDA (Retired), Laurel, Maryland, USA

Address correspondence to Kenneth S. Macklin, ksm137@msstate.edu.

The authors declare no conflict of interest.

See the funding table on p. 18.

breeders, and broilers). Similarly, the phylogenetic analysis showed strong genetic relationship among strains isolated from different locations within the same stage and between different stages. The results show complex diversity of *Salmonella* serotypes along the chain and the possibility of multiple critical points for the entry of pathogen into the broiler complex and contaminate the final raw product at the processing plant. Furthermore, improper cooking or handling of contaminated raw chicken meat and meat products with *Salmonella* and other zoonotic pathogens can potentially cause foodborne illness in humans.

**KEYWORDS** *Salmonella*, chickens, production farms, facilities, sample types, prevalence

In the United States, poultry meat consumption is considerably higher than beef or pork alone (1) and most of the broiler companies are vertically integrated (2). Despite the 50% decrease in the proportion of chicken samples positive for *Salmonella* during the years from 2017 to 2021, the incidence of human *Salmonella* infections has remained consistent over the last two decades (3). Moreover, the third component of USDA proposed regulatory framework for reducing *Salmonella* in poultry is more focused on quantitative infective dose and prevalence of serotypes that cause people to become sick rather than the overall *Salmonella* serotypes (3). Although several efforts have been made to better understand the ecology and epidemiology of the pathogen, its frequent incidence and outbreaks in food products necessitate development of improved control measures against the pathogen.

*Salmonella* is a Gram-negative, facultatively anaerobic, non-spore-forming, rod-shaped bacteria of Enterobacteriaceae family (4). It is considered as the leading bacterial pathogen causing foodborne illness in humans worldwide (5, 6). The Centers for Disease Control and Prevention (CDC) estimates 1.35 million infections, 26,500 hospitalizations, and 420 deaths due to *Salmonella* every year in the United States (7). Non-Typhoidal *Salmonella* infections may cause acute gastroenteritis, fever, chills, nausea, and vomiting. They are usually self-limiting and subside within a week without any medications, but sometimes, the infections can be fatal, especially in young, immunocompromised, elderly people, and people with medications like antacids (8, 9). In addition, the annual economic burden of foodborne illness in the United States due to *Salmonella* contamination from chicken products accounts for $2.8 billion (10).

*Salmonella* is usually present in the gastrointestinal tract of domestic and wild animals including humans and birds. Reused litter, water, feed, dust, infected personnel, and contaminated equipment can act as reservoirs for *Salmonella* in chicken houses (4, 11, 12) since the microorganism can survive desiccation and can persist for years in dry environments and foods (13, 14). Horizontal transmission of the pathogen can occur across the birds within the flock by contaminated food and water or via aerosol (15, 16). Alternately, *Salmonella* infected poultry serves as a reservoir for infecting healthy humans via consumption or handling of contaminated undercooked or raw chicken meat and products (17, 18) (Fig. 1). The consumption of contaminated chicken and turkey meat accounts for 23% of foodborne *Salmonella* illnesses (19). Therefore, the reduction or elimination of *Salmonella* along poultry food chain will have a significant impact on food safety and public health.

Most of the previous studies (20–23) focused on sampling of broiler grow-out farms to raw products in the processing plant for *Salmonella* detection as these stages have direct implications with foodborne illness. A few studies (24–28) also considered breeder flocks and hatcheries for the isolation of bacteria. However, studies that involve in-depth understanding of genetic relatedness and tracking of *Salmonella* strains along the poultry chain using whole-genome sequencing (WGS) with single nucleotide polymorphism (SNP) based analysis are rarely reported (29–31). For the control of *Salmonella*, the major challenges are to reduce the risk of introduction or spread of *Salmonella* in flocks, to minimize the persistence of bacterial strains within the broiler complex, and to achieve the USDA FSIS performance standards at processing plants. The objective of this

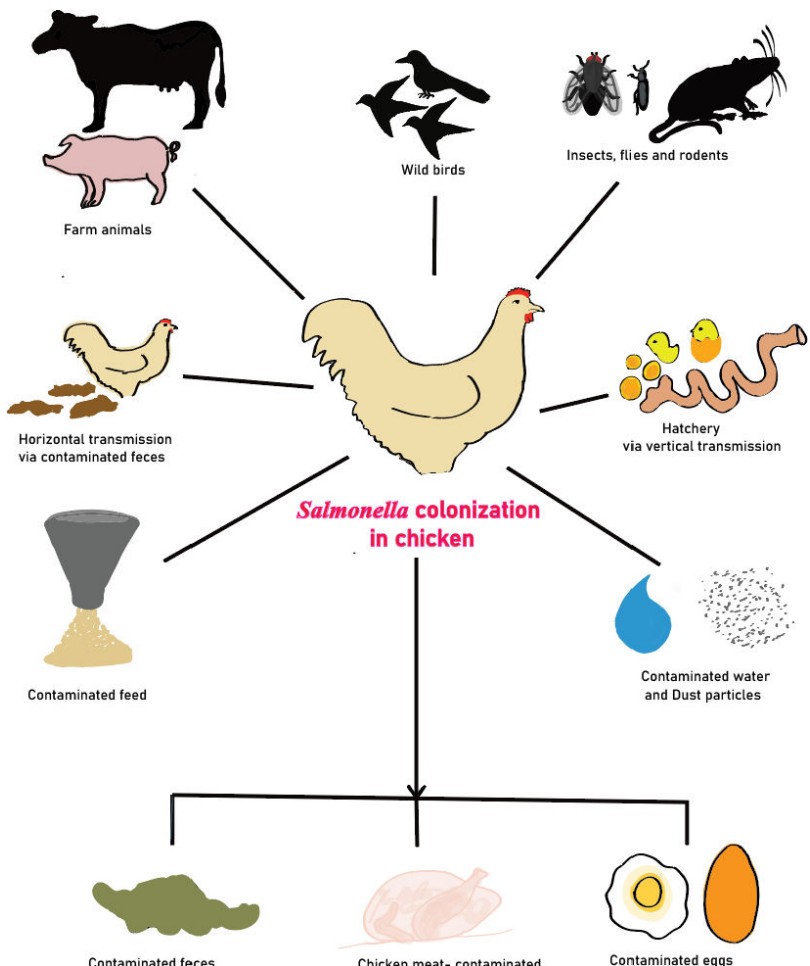

**FIG 1**  Potential sources of *Salmonella* colonization in chicken. This figure displays potential sources of *Salmonella* inside and outside the poultry houses and facilities which can potentially cause its colonization inside the birds (figure drawn with Adobe Illustrator).

longitudinal study was to determine the prevalence, critical entry points, phylogenetic relatedness, and movement patterns of *Salmonella* along the different stages of integrated broiler production complexes to improve pathogen control strategies.

## RESULTS

*Salmonella* was detected in all stages and almost all sample types. Out of 38 different production houses, 47% (18 out of 38) of houses tested MDS positive for *Salmonella*, while 42% (16 out of 38) of houses tested *Salmonella* positive by both MDS and culture (Tables 1 and 2). Similarly, 21% (227 out of 1,071) of the samples were positive by MDS, while 18% (192 out of 1,071) were positive by both MDS and culture, which were further confirmed with PCR assay (Table 3). Interestingly, 100% of breeder farms (five farms or nine houses) tested *Salmonella* positive on both MDS and culture. Based on culture results, 8% (29 out of 375) of samples tested positive from parent farms, 2% (7 out of 302) tested positive from broiler farms, 25% (16 out of 63) tested positive from hatchery, 40% (37 out of 98) tested positive from transport, and 43% (103 out of 238) tested positive from the processing plant (Table 3). The detailed culture positive result along with the stages and sample types is available in Table S1.

**TABLE 1** MDS and culture results of production farms and their corresponding houses[a]

| Farm type | Farm name | House | Age | Flock density (birds/house) | MDS results | Culture results |
|---|---|---|---|---|---|---|
| Pullet farms | Farm A | House 5 | 14 weeks | 14,000 | – | – |
| | | House 6 | 14 weeks | 14,000 | – | – |
| | Farm B | House 1 | 10 weeks | 9,000 | – | – |
| | | House 5 | 12 weeks | 14,000 | – | – |
| | Farm C | House 1 | 11 days | 15,000 | – | – |
| | | House 4 | 11 days | 15,000 | + | + |
| | Farm D | House 3 | 7 weeks | 9,200 | + | + |
| | | House 6 | 7 weeks | 9,200 | + | + |
| Breeder farms | Farm A | House 1 | 46 weeks | 11,500 | + | + |
| | | House 2 | 46 weeks | 11,500 | + | + |
| | Farm B | House 1 | 31 weeks | 11,500 | + | + |
| | | House 2 | 31 weeks | 11,500 | + | + |
| | Farm C | House 1 | 45 weeks | 8,200 | + | + |
| | | House 3 | 45 weeks | 11,000 | + | + |
| | Farm D | House 3 | 59 weeks | 8,200 | + | + |
| | | House 4 | 58 weeks | 8,200 | + | + |
| | Farm E | House 1 | 55 weeks | 12,000 | + | + |
| | | House 3 | 52 weeks | 12,000 | – | – |
| Broiler farms | Farm A | House 1 | 47 days | 21,500 | – | – |
| | | House 4 | 47 days | 21,500 | – | – |
| | Farm B | House 1 | 15 days | 28,500 | + | + |
| | | House 6 | 15 days | 28,500 | – | – |
| | Farm C | House 1 | 21 days | 41,000 | + | – |
| | | House 3 | 21 days | 41,000 | – | – |
| | Farm D | House 1 | 35 days | 41,000 | – | – |
| | | House 2 | 35 days | 41,000 | – | – |
| | Farm E | House 1 | 26 days | 42,000 | + | + |
| | | House 3 | 26 days | 42,000 | – | – |
| | Farm F | House 1 | 33 days | 21,000 | – | – |
| | | House 6 | 33 days | 21,000 | – | – |
| | Farm G | House 1 | 16 days | 28,500 | – | – |
| | | House 4 | 16 days | 28,500 | – | – |
| | Farm H | House 1 | 14 days | 28,500 | + | + |
| | | House 4 | 14 days | 28,500 | – | – |
| | Farm I | House 1 | 19 days | 57,000 | – | – |
| | | House 2 | 19 days | 57,000 | + | + |
| | Farm J | House 1 | 21 days | 28,500 | – | – |
| | | House 4 | 21 days | 28,500 | + | – |

[a]"+", positive; "-", negative.

## *Salmonella* prevalence among different stages and sample types

The odds of *Salmonella* detection in hatchery samples were four times (1.85–7.55; 95% CLs) as likely as its detection in production farms' samples ($P = 0.0011$) (Table 4). Similarly, the odds of *Salmonella* detection in transport samples were 10 times (4.63–82.12) as likely as its detection in production farms' samples ($P < 0.0001$). Moreover, the odds of *Salmonella* detection in processing plant samples were 33 times (14.46–82.12) as likely as its detection in production farms' samples ($P < 0.0001$).

Among the production farms, the odds of *Salmonella* detection in breeder farms' samples were five times (2.25–13.32) as likely as its detection in broiler farms' samples ($P = 0.0007$). Similarly, the odds of *Salmonella* detection in boot swabs were three times (0.39–69.26) as likely as its detection in litter grab samples ($P = 0.319$). However, there were no significant differences among different sample types for *Salmonella* detection

**TABLE 2** MDS and culture results of three follow-up sampling visits from broiler farms to transport to processing plant

| Follow-up visits | Farm/facilities | Farm name | House | MDS results | Culture results |
|---|---|---|---|---|---|
| First follow-up visit | Broiler farm | Farm B | House 1 | − | − |
| | Transport trailers | | | + | + |
| | Processing plant | | | + | + |
| Second follow-up visit | Broiler farm | Farm J | House 4 | − | − |
| | Transport trailers | | | + | + |
| | Processing plant | | | + | + |
| Third follow-up visit | Broiler farm | Farm E | House 1 | − | − |
| | Transport trailers | | | + | + |
| | Processing plant | | | + | + |

*a*"+", positive; "-", negative.

within the production farms. Moreover, the odds of *Salmonella* detection in outdoor farm environmental samples were 1.23 times (0.23–8.48) as likely as its detection in indoor farm environmental samples ($P = 0.8192$). The odds of *Salmonella* detection in summer season were 1.57 times (0.43–6.37) as likely as its detection in spring season ($P = 0.0573$). However, the results were not statistically significant ($P > 0.05$) (Table 4).

Similarly, among transport and processing plant samples, the odds of *Salmonella* detection in processing plant samples were four times (1.87–9.74) as likely as its detection in transport trucks' samples ($P = 0.0007$) (Table 5). In addition, the odds of *Salmonella* detection in post-pick whole carcass rinses were 105 times (37.39–350.77) as likely as its detection in post-chill whole carcass rinses in the processing plant ($P < 0.0001$). Similarly, the odds of *Salmonella* detection in sponge-stick swabs and fecal samples were significantly higher ($P ≤ 0.003$) as compared to its detection in post-chill whole carcass rinses.

## Serotypes identification and critical entry points of *Salmonella*

The serotypes identified in this study were *S.* Kentucky (64%), *S.* Enteritidis (9%), *S.* Typhimurium (3%), *S.* Johannesburg (3%), *S.* Montevideo (3%), *S.* Mbandaka (3%), *S.* Newport (2%), *S.* Senftenberg (1%), *S.* Inverness (1%), *S.* Ohio (1%), *S.* Uganda (2%), and N/A (9:z29:-) (2%) (Table 6). Interestingly, the serotypes of foodborne importance like *S.* Enteritidis, *S.* Typhimurium, and *S.* Newport were identified in breeder farms and only *S.* Enteritidis was identified in hatchery, broiler farms, and processing plants. However, these serotypes were not identified in pullet farms and transport.

Based on PCR-confirmed culture results, *Salmonella* was isolated from all stages and 11 different sample types of an integrated broiler complex. Interestingly, this study

**TABLE 3** Overall prevalence of *Salmonella* based on MDS and culture results along with the serotypes identified in this study in each location

| Stages | MDS positive | Culture positive | Serotypes |
|---|---|---|---|
| Pullet | 7/177 (Swabs, fly paper, water samples, miscellaneous) | 4/177 (Swabs, water sample, miscellaneous) | Mbandaka, Kentucky |
| Breeder | 33/198 (Boot swabs, swabs, beetle traps, fly papers, litter grab, soil, water samples, miscellaneous) | 25/198 (Boot swabs, swabs, beetle traps, fly papers, litter grab, soil, water samples, miscellaneous) | Inverness, Newport, Kentucky, Senftenberg, Enteritidis, Typhimurium |
| Hatchery | 28/63 (Boot swabs, swabs, fly paper, miscellaneous) | 16/63 (Boot swabs, swabs, fly paper, miscellaneous) | Johannesburg, Ohio, Kentucky, Enteritidis, N/A (9:z29:-) |
| Broiler | 10/302 (Swabs, feed, water samples, miscellaneous) | 7/302 (Swabs, water samples, miscellaneous) | Enteritidis, Montevideo, Kentucky |
| Transport | 42/93 (Swabs, feces) | 37/93 (Swabs, feces) | Kentucky, Alachua |
| Processing plant | 107/238 (Swabs, feces, carcass rinses) | 103/238 (Swabs, feces, carcass rinses) | Kentucky, Enteritidis, Uganda, Alachua |
| Total | 227/1,071 (21.20%) | 192/1,071 (17.93%) | |

**TABLE 4** Comparison of variables within using generalized linear modeling for categories: stages of production, different farms, sample types, environments, and seasons

| | Positive/total[a] | Odds ratio (OR) | Lower CL | Upper CL | Standard error | P value (Tukey) |
|---|---|---|---|---|---|---|
| **Stages** | | | | | | |
| Production farms | 36/640[C] | | | Reference group | | |
| Processing plant | 103/238[A] | 33.228 | 14.455 | 82.117 | 0.442 | <0.0001 |
| Transport | 37/93[B] | 9.568 | 4.632 | 20.526 | 0.379 | <0.0001 |
| Hatchery | 16/63[B] | 3.794 | 1.853 | 7.548 | 0.357 | 0.0011 |
| **Production farms** | | | | | | |
| Broiler | 7/280[B] | | | Reference group | | |
| Breeder | 25/189[A] | 5.141 | 2.245 | 13.318 | 0.447 | 0.0007 |
| Pullet | 4/171[B] | 0.942 | 0.240 | 3.236 | 0.645 | 0.9953 |
| **Sample types** | | | | | | |
| Litter grab | 1/41 | | | Reference group | | |
| Beetle trap | 2/73 | 1.093 | 0.097 | 24.560 | 1.263 | 1.0000 |
| Boot swab | 3/41 | 3.284 | 0.387 | 69.260 | 1.193 | 0.9751 |
| Fly paper | 2/157 | 0.482 | 0.032 | 11.941 | 1.361 | 0.9995 |
| Miscellaneous | 7/104 | 1.920 | 0.142 | 49.784 | 1.382 | 0.9998 |
| Soil | 2/38 | 1.774 | 0.085 | 57.237 | 1.557 | 1.0000 |
| Sponge stick swabs | 12/95 | 4.560 | 0.819 | 85.603 | 1.072 | 0.8504 |
| Water drainage | 7/91 | 2.444 | 0.170 | 67.434 | 1.423 | 0.9985 |
| **Environment** | | | | | | |
| Indoor | 19/324 | | | Reference group | | |
| Outdoor | 17/316 | 1.233 | 0.226 | 8.476 | 0.912 | 0.8192 |
| **Season** | | | | | | |
| Spring | 4/171 | | | Reference group | | |
| Summer | 26/302 | 3.555 | 1.330 | 12.348 | 0.554 | 0.0572 |
| Fall | 6/167 | 1.566 | 0.426 | 6.374 | 0.670 | 0.7813 |

[a]Data were analyzed using generalized linear modeling for binomial distribution. Means not sharing a common letter were significantly different ($P < 0.05$).

suggests multiple critical entry points and complex diversity of *Salmonella* serotypes along the broiler production chain. This is because the *Salmonella* serotypes present in the latter stages were not present in the upstream stages. Alternatively, most of the serotypes present in upstream stages were not found in transport and processing plants along the chain (Fig. 2). However, in our study, the environment of facilities such as processing plants, transport, and hatchery were most likely to be contaminated with *Salmonella*. There might be the possibility of pathogen persistence in these environments as reservoirs and disseminate bacterium along the food chain. In addition, 88% (56 out of 64) of *Salmonella*-positive post-pick whole carcass rinses were significantly

**TABLE 5** Comparison of variables within transport and processing plant using generalized linear modeling for categories: stages and sample types

| | Positive/total[a] | Odds ratio (OR) | Lower CL | Upper CL | Standard error | P value (Tukey) |
|---|---|---|---|---|---|---|
| **Stages** | | | | | | |
| Transport | 37/93[B] | Reference group | | | | |
| Processing plant | 103/216[A] | 4.137 | 1.874 | 9.742 | 0.418 | 0.0007 |
| **Sample types** | | | | | | |
| Carcass rinses (post-chill) | 6/96[C] | Reference group | | | | |
| Carcass rinses (post-pick) | 56/64[A] | 105.004 | 37.385 | 350.769 | 0.566 | <0.0001 |
| Sponge-Stick swabs | 69/101[A] | 80.656 | 28.673 | 265.559 | 0.563 | <0.0001 |
| Fecal sample | 9/48[B] | 8.616 | 2.619 | 30.250 | 0.616 | 0.0027 |

[a]Data were analyzed using generalized linear modeling for binomial distribution. Means not sharing a common letter were significantly different ($P < 0.05$).

**TABLE 6** Number of serotypes identified in this study and their abundance in each sample type considered

| Serotypes | Sample types | | | | | | | | | | | Total |
|---|---|---|---|---|---|---|---|---|---|---|---|---|
| | Boot swabs | Swabs | Beetle traps | Fly papers | Litter grab | Soil | Water | Feces | Rinses post-pick | Rinses post-chill | Miscellaneous | |
| Kentucky | 5 | 27 | 2 | 3 | 1 | 0 | 2 | 5 | 9 | 5 | 4 | 64 |
| Enteritidis | 1 | 4 | | 1 | | | 1 | | | | 2 | 9 |
| Alachua | | 5 | | | | | | 1 | | | | 6 |
| Typhimurium | | | | | | 1 | 2 | | | | | 3 |
| Johannesburg | | 1 | | | | | | | | | 2 | 3 |
| Montevideo | | | | | | | 2 | | | | 1 | 3 |
| Mbandaka | | 2 | | | | | | | | | 1 | 3 |
| Newport | | | | | | | | | | | 2 | 2 |
| Uganda | | 2 | | | | | | | | | | 2 |
| Senftenberg | | 1 | | | | | | | | | | 1 |
| Inverness | | | | | | 1 | | | | | | 1 |
| Ohio | | 1 | | | | | | | | | | 1 |
| N/A | | | | 2 | | | | | | | | 2 |
| Total | 6 | 43 | 2 | 6 | 1 | 2 | 7 | 6 | 9 | 5 | 12 | 100 |

reduced to 6% (6 out of 96) of *Salmonella*-positive post-chill carcass rinses which confirms the high effectiveness of chilling (with antimicrobials) step in the processing plant.

## Phylogenetic analyses

The core genome-based phylogeny of *Salmonella* strains from this study with LT2 reference strain is presented in Fig. 3. Interestingly, the phylogenetic analysis showed strong genetic relationship among bacterial strains isolated from different stages within an integrated broiler complex. The *S*. Enteritidis strains from different stages of hatchery (MACK4, MACK6, and MACK13) and from broiler farms (MACK105) showed high genetic relatedness with SNPs ≤ 6 (Fig. 4A). Similarly, the *S*. Alachua strains isolated from the live bird hang area in the processing plant (MACK58) and from swabbing transport cages' floor (MACK63, MACK75, MACK76, MACK77, and MACK78) showed close genetic

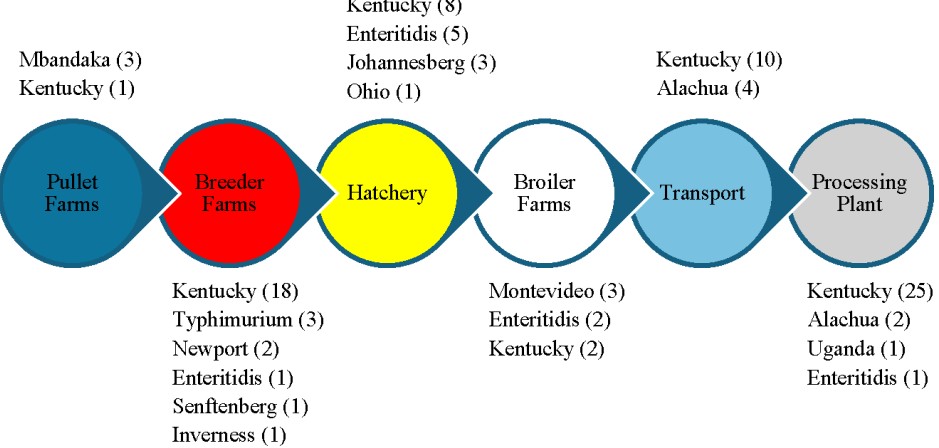

**FIG 2** Serotypes distribution along different stages of broiler production chain. The figure displays different stages of an integrated broiler complex, and the serotypes of 100 representative *Salmonella* isolates identified in this study. The number inside the bracket represents the number of samples with serotypes identified from each stage. It can be observed from the figure that all the serotypes present in the former stages of poultry chain did not make up to processing plant. Alternatively, the serotypes that were present in the latter stages of chain were not necessarily present in the previous stages. The results support multiple critical entry points of bacteria along the broiler production chain.

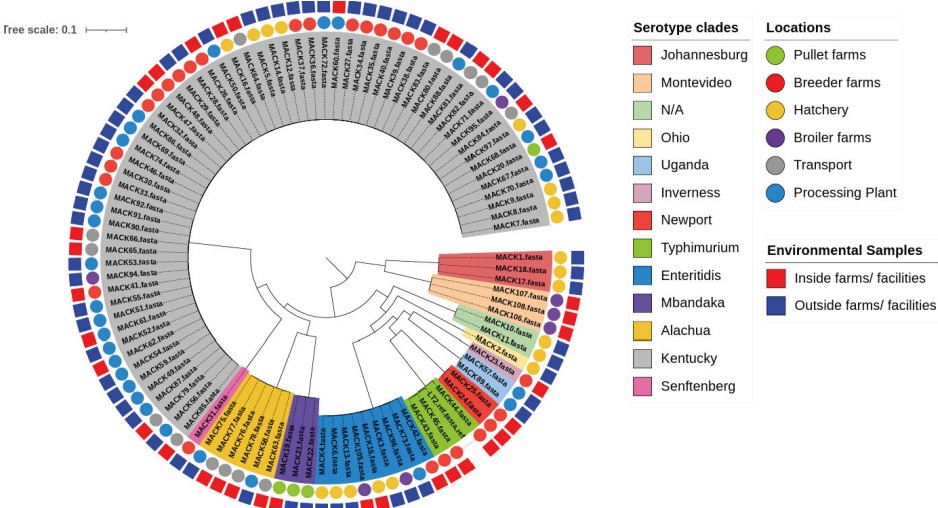

**FIG 3** Core genome-based phylogeny of 100 representative *Salmonella* isolated from this study with reference genome being *S. Typhimurium* LT2 strain from NCBI. The figure displays three different circles. Inner circle represents clades of 13 different serotypes identified in this study represented by different colors. The middle circle represents different stages of broiler production systems, while the outermost circle represents inside or outside environmental samples from farms and facilities.

relatedness with SNPs ≤ 5 (Fig. 4B). Similarly, the *S.* Mbandaka strains isolated from outside and inside environments of pullet houses (MACK19. MACK21, and MACK22) showed a high genetic relationship with SNPs ≤ 6 (Fig. 4E). Finally, the 64 *S.* Kentucky strains isolated from this study showed 13 closely related clusters which strongly showed the genetic relationships (SNPs ≤ 12) between strains isolated from inside and outside environments of processing plant, transport, breeder, broiler, and pullet houses (Fig. 5). The phylogenetic results suggest the possibility of transmission of the same *Salmonella* strains between outside and inside surroundings of farms/facilities and between the different stages of the broiler production chain (Fig. 5 and 6).

The *Salmonella* strains of each serotype from this study had separate clusters as compared to NCBI strains. When compared with NCBI strains, the *S.* Uganda, *S.* Mbandaka, *S.* Typhimurium, and *S.* Alachua strains from this study had SNPs > 200 which suggests distant genetic relatedness among bacterial strains. However, *S.* Enteritidis strains from this study had SNPs between 29 and 65 with NCBI strains. Moreover, NCBI strains isolated from broiler environments and stool samples in Canada had SNPs between 29 and 31 when compared with *S.* Enteritis strains (MACK4, MACK6, MACK13, and MACK105). Similarly, when compared with NCBI strains, the *S.* Montevideo, *S.* Johannesberg, *S.* Newport, *S.* Inverness, and *S.* Senftenberg strains from this study had SNPs > 70 which suggests distant genetic relatedness among these bacterial strains.

## DISCUSSION

The results showed that *Salmonella* was present in the environment of all production farms and facilities of an integrated broiler complex. Similar to the findings of this study, previous studies (25–28, 32–35) also reported *Salmonella* contamination in multiplier/breeder farms, broiler farms, hatcheries, transport trucks, and processing plants of integrated broiler complexes. The poultry house environments such as reused litter, equipment, dust particles, contaminated protein-rich feed and water could serve as an important reservoir for *Salmonella* colonization (4, 11, 12). In this study, the odds of *Salmonella* detection were significantly more likely in hatcheries, transport trucks, and processing plant samples as compared to its detection in production farms. Similar trends were observed in the previous study (35); however, the likelihood of *Salmonella* detection in such facilities was not statistically significant as compared to production farms' samples in that study. The higher prevalence of *Salmonella* in transport and

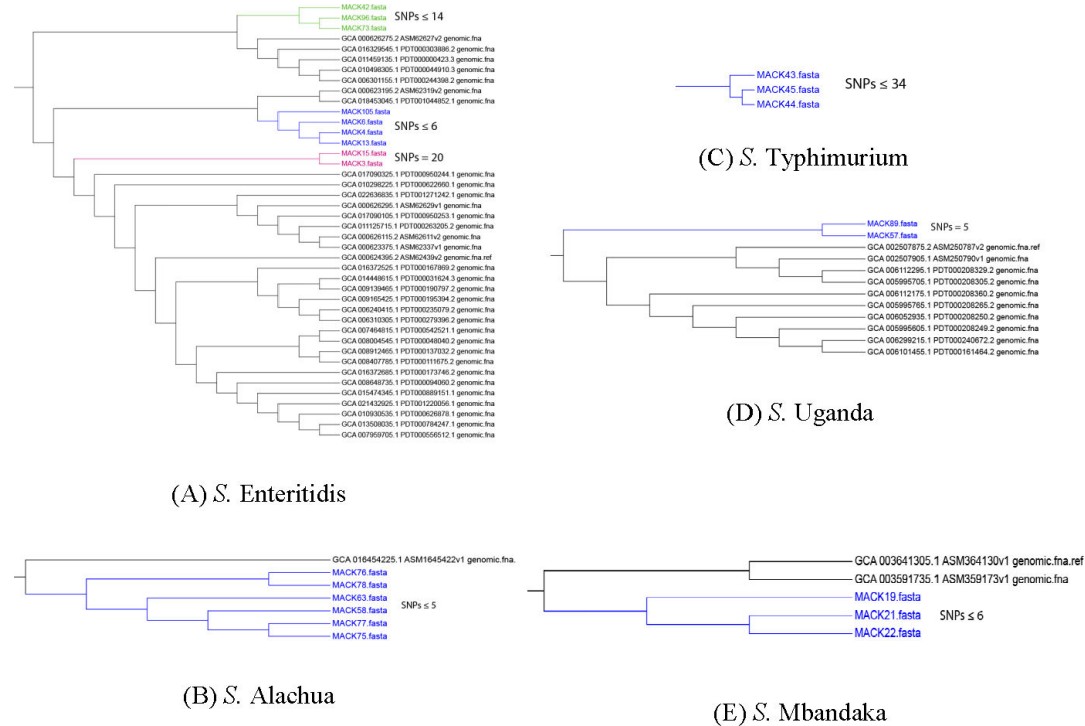

**FIG 4** Phylogenetic trees of (A) *S.* Enteritidis, (B) *S.* Alachua, (C) *S.* Typhimurium, (D) *S.* Uganda, and (E) *S.* Mbandaka strains showing different clusters with number of SNPs in each cluster determined by CSI phylogeny service of CGE website. Among nine *S.* Enteritidis strains isolated from this study, there were three distinct clusters. Cluster A consisted of strains isolated from composite eggs samples (MACK4), clean egg buggies swabs (MACK6), hatcher boot swabs (MACK13) in hatchery and from ants (MACK105) in broiler farms with SNPs ≤ 6. Cluster B consisted of strains isolated from swab of egg holders at Embrex machine (MACK3) and fly paper near dumpster (MACK15) in the hatchery with SNPs 20. Cluster C consisted of strains isolated from water drainage outside the breeder house (MACK42), swab from moving belt (MACK73) at the live birds hang area in the processing plant, and from fan exhaust swab (MACK96) inside the broiler house with SNPs ≤ 14. The SNPs among *S.* Alachua strains isolated from swabbing the moving conveyor belt (MACK58), fresh feces (MACK75) at the live birds hang area in the processing plant, and from swabbing transport cages' floor (MACK63, MACK76, MACK77, and MACK78) were ≤5. The SNPs among *S.* Typhimurium strains isolated from a soil sample (MACK43) and water drainage samples (MACK44 and MACK45) outside the breeder house were ≤34. The SNPs among *S.* Uganda strains from swabbing live birds hang area conveyor belt (MACK57 and MACK89) were 5. The SNPs among *S.* Mbandaka strains isolated from unknown feces outside pullet house (MACK 19) and from swabbing fan exhausts inside pullet house (MACK21 and MACK22) were ≤6. Similarly, the SNPs between *S.* Newport strains isolated from rodents feces outside the breeder houses (MACK24 and MACK25) were 2. In addition, the SNPs between *S.* Johannesburg strains from incubator swab in the hatchery (MACK1) and from rodent intestines (MACK17 and MACK18) were 48.

processing plant indicates that the bacterial contamination in broiler grow-out flocks can be further increased during transport and slaughter activities due to cross-contamination and stress arousal activities during the handling of birds. *Salmonella* can survive and persist in dry environments for years (14). Medina-Santana et al. also reported the persistence of *S.* Infantis clones in all levels of production chain despite sanitation, ultimately contaminating carcasses in the slaughter facilities (31). A similar finding was reported in a meta-analysis by Wang et al. (12) in which hatchery was the most significant contributor to *Salmonella* prevalence with 48.5% in poultry live production system. However, the results of this meta-analysis were based only on pre-harvest live poultry production system, and it did not consider transport trucks and processing plants in the study. Therefore, the environment of facilities may serve as reservoirs for bacteria to spread and contaminate the chicken product along the chain. Strict sanitation and disinfection procedures may be essential to reduce or eliminate such bacteria from these facilities/farms.

In this study, we identified 12 different *Salmonella* serotypes with predominant serotype being *S.* Kentucky followed by *S.* Enteritidis, *S.* Alachua, *S.* Typhimurium, *S.* Johannesburg, *S.* Montevideo, *S.* Mbandaka, *S.* Newport, *S.* Senftenberg, *S.* Inverness,

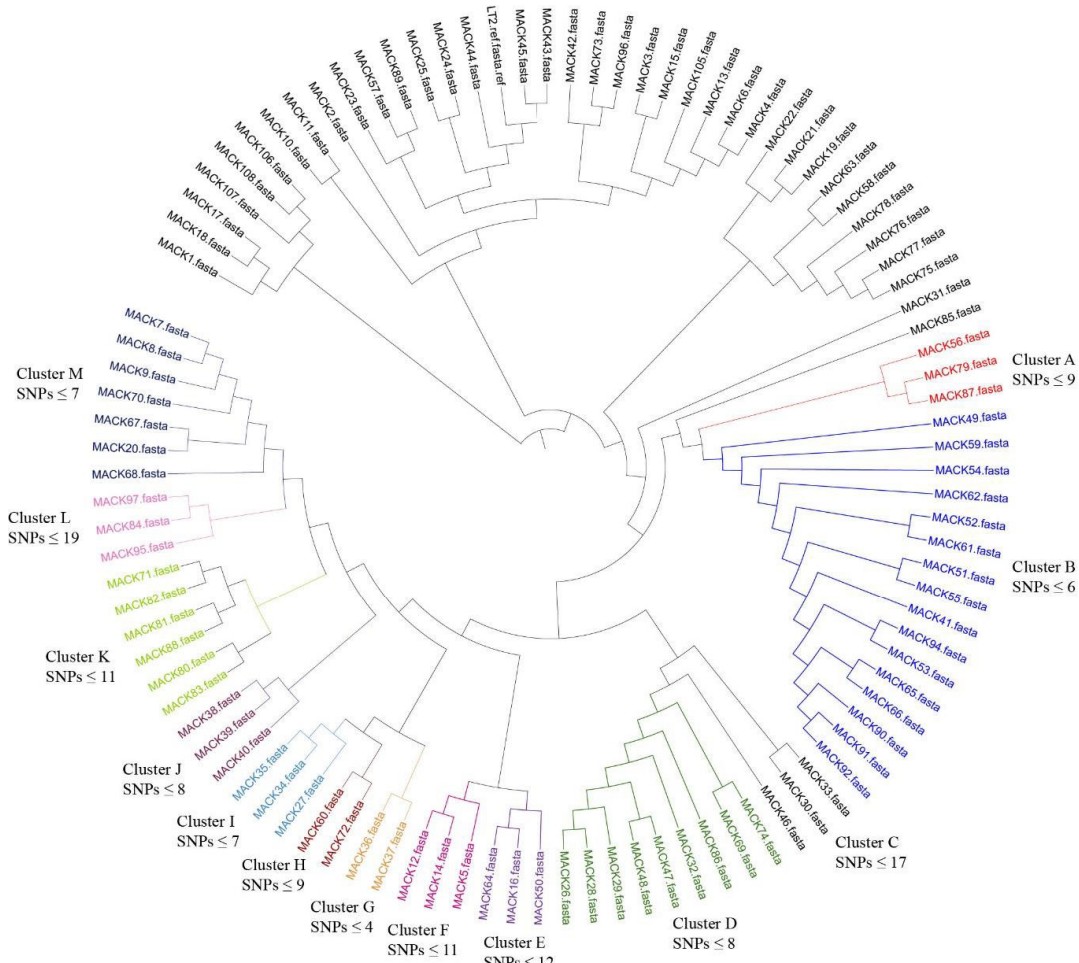

**FIG 5** Phylogenetic tree of *S.* Kentucky strains isolated from this study showing 13 different clusters from A to M along with the number of SNPs determined for each cluster by using CSI phylogeny service of CGE website. Cluster A consisted of strains isolated by swabbing the moving conveyor belt of live birds hang area (MACK56 and MACK87) and transport cages' floor (MACK79) with SNPs 9. Cluster B consists of strains isolated from post-pick carcass rinses (MACK49, MACK51, MACK52, MACK90, MACK91, and MACK92), post-chill carcass rinses (MACK53, MACK54, and MACK55), from fresh feces at live birds hanging area (MACK59), by swabbing the moving conveyor belt in the live birds unloading area (MACK61 and MACK62) in the processing plant, by swabbing the floor of transport cages' floor (MACK65 and MACK66), by swabbing fan exhausts (MACK94) in the broiler farms, from fly paper (MACK41) outside the breeder house with SNPs < 6. Cluster C consisted of strains isolated from beetle trap (but no beetles, only litter) (MACK30 and MACK33) and by swabbing moving egg collection conveyor belt (MACK46) in the breeder but these strains have SNPs ≤ 17.Cluster D consisted of strains isolated by swabbing moving conveyor belt at live birds hang area (MACK74), from post-pick carcass rinses (MACK69), from fresh feces collected from live birds hang area (MACK86) in the processing plant, from litter grab (MACK29) and fly paper (MACK32) inside the breeder house and from miscellaneous samples like wild bird or rodent feces (MACK47 and MACK48) outside the breeder houses with SNPs ≤ 8. Cluster E consisted of strains isolated from post-pick carcass rinses (MACK50) in the processing plant, from fly paper (MACK16) near dumpster outside the hatchery and by swabbing transport cages' floor (MACK64) with SNPs ≤ 12. Cluster F consisted of strains isolated from discarded composite egg samples near Embrex machine (MACK5) and boot swabs (MACK12 and MACK14) from setter room and corridor or hallway in the hatchery with SNPs ≤ 11. Cluster G consisted of strains isolated from boot swab (MACK36) and by swabbing fan exhausts (MACK37) inside the breeder house with SNPs ≤ 4. Cluster H consisted of strains isolated by swabbing the moving conveyor belt (MACK60) at live birds loading area and by swabbing moving conveyor belt (MACK72) at live birds hang area in the processing plant with SNPs 9. Cluster I consisted of strains isolated by swabbing fan exhausts (MACK27), nest boxes (MACK34), and egg collection moving conveyor belt (MACK35) inside the breeder house with SNPs 7. Cluster J consisted of strains isolated from boot swab (MACK38), by swabbing nest boxes (MACK39) and from dirty egg wipe paper (MACK40) inside the breeder house with SNPs 8. Cluster K consisted of strains isolated from post-chill carcass rinses (MACK71) in the processing plant, by swabbing transport cages' floor (MACK80, MACK81, and MACK82), from feces in transport cages and truck trailer floor (MACK83) and by swabbing moving conveyor belt at live birds hang area (MACK88) in the processing plant with SNPs ≤ 11. Cluster L consisted of strains isolated from feces from transport cages and truck trailer floor (MACK84) and by swabbing fan exhausts (MACK95) inside the broiler house, and from boot swab (MACK97) in separator room at hatchery with SNPs ≤ 19. Cluster M consists of strains isolated from post-pick carcass rinses (MACK67 and MACK68), post-chill carcass rinses (MACK70) in the processing plant, by swabbing of dirty chick trays (MACK7, MACK8, and MACK9) just delivered from broiler farms in the hatchery and from water drainage outside the pullet house with SNPs ≤ 7.

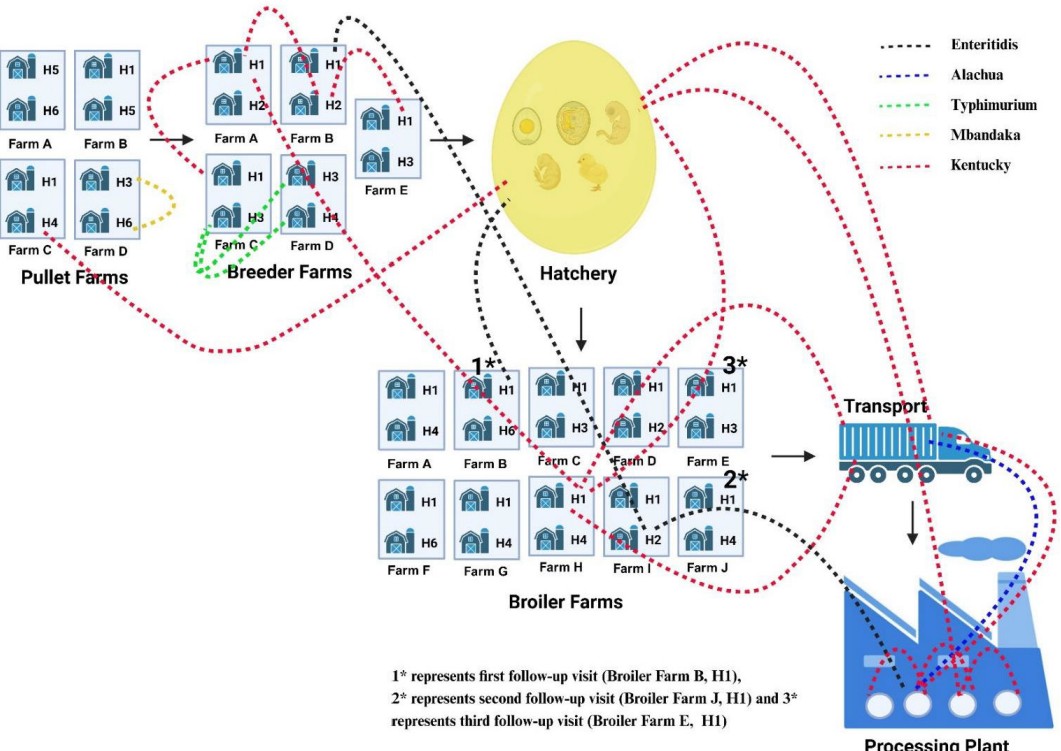

**FIG 6** Movement patterns of *Salmonella* strains. The figure displays the different stages of an integrated broiler complex that were sampled during the study. The stages include 8 pullet houses (4 pullet farms), 10 breeder houses (5 breeder farms), hatcheries, 20 broiler houses (10 broiler farms), transport, and processing plants. The dotted lines represented by different colors such as black (Enteritidis), blue (Alachua), green (Typhimurium), yellow (Mbandaka), and red (Kentucky) display the possibility of movement of same *Salmonella* strains along different locations within same stage or between different stages of an integrated broiler complex.

*S.* Ohio, and *S.* Uganda. The nature of serotype distribution along the broiler production chain was complex and diverse. Most of the serotypes that were present in the former stages along the chain did not make it up to the latter stages. Alternately, the strains that were not recovered from production farms appeared in the processing plant and vice-versa which suggests the persistence of this organism in transport and the processing plant. This persistence can potentially cross-contaminate incoming birds which is consistent with the findings of Liljebjelke et al. (27). A similar pattern of *Salmonella* serotype distribution was reported in turkey flocks in the United States (36). Our study indicated multiple entry points of bacteria along the broiler production chain which may ultimately lead to higher contamination at the processing plant. Similar findings of multiple entry points were reported by Siceloff et al. in which *S.* Kentucky was reported as predominant serotype in breeder flocks, but not in the processing plant (37). More specifically, only one swab sample from the live birds' hang area in the processing plant was contaminated by *S.* Enteritidis. *S.* Kentucky is most frequently isolated from poultry in the United States (37) which supports the findings of our study. The reason may be due to its unique selective adaptational growth that could outcompete other serotypes because Kentucky strains were reported to persist longer and at higher cell densities in the ceca of birds as compared to Typhimurium strains (38). Moreover, stationary sigma factor *rpoS* regulatory genes associated with stress and metabolic responses were highly expressed in Kentucky strains as compared to Typhimurium strains (38). Similarly, Guillén et al. reported that Kentucky strains showed lower acid resistance and higher heat resistance, and similar oxidative resistance as compared to Enteritidis strains (39). In addition, Kentucky strains might have outcompeted other serotypes present in a single sample during selective enrichment with RV or TT broth (40). Another possibility can be

that the pre-enrichment promoted abundant growth of Kentucky serotype which masked the detection of other serotypes during the conventional culture method. Therefore, deep serotyping of *Salmonella* isolates that could detect multiple serotypes from a single sample could help to further elucidate the tracking and transmission of pathogens along the broiler production chain. Moreover, future studies may be directed to metagenomic analyses which can provide detailed information about entire microbes present in a sample.

Among 12 different sample types, boot swabs, sponge-stick swabs, post-pick whole carcass rinses, water samples, and miscellaneous samples were found to be useful environmental sample types for *Salmonella* detection. Similar to the finding of this study, boot socks were reported as the most sensitive sample type for detection of *Salmonella* in farms and the most identified *Salmonella* serotypes were *S*. Kentucky and *S*. Enteritidis (22). Similarly, higher recovery of *Salmonella* was reported from fly papers, boot swabs, and outside dirt near the entry doors (41). Moreover, Trimble et al. reported *Salmonella* contamination in soil, compost, and wastewater from a small-scale pasture-raised broiler farm (42). In this study, sponge-stick swab samples were used to consider dust samples from the walls and fan exhausts inside the poultry houses. *Salmonella* can exist in aerosolized form within dust particles, and swabbing dust from poultry houses and facilities may have a high likelihood of detecting pathogens (16, 27, 43, 44). Miscellaneous samples like rodents' intestines, wild bird feces, or unknown feces in the outside environment of poultry houses also pose as potential reservoirs of *Salmonella* (43, 45) which can potentially transmit pathogens inside the poultry houses and potentially to birds. *S*. Johannesburg and *S*. Newport recovered from rodents' intestines in hatchery and unknown feces in breeder farms, respectively, in this study. However, those serotypes were not recovered from subsequent stages of broiler production systems.

The phylogenetic analysis shows strong genetic relatedness among the bacterial strains isolated from different stages and sample types in this study, similar to the findings of our previous study (35). The different clusters of *S*. Kentucky, *S*. Enteritidis, *S*. Alachua, and *S*. Mbandaka strains indicate the possibility of pathogen movement along different locations within the same stage and between different stages of broiler production systems. We observed that within the different production farms, the odds of *Salmonella* detection were significantly higher in breeder farms as compared to broiler and pullet farms. Since the organism shows vertical nature of transmission, the parent breeder birds could serve as reservoir for pathogen and disseminate it along the subsequent stages in broiler production chain. Similar to the findings of this study, an 18-month longitudinal study in a vertically integrated poultry operation in Australia reported parent breeder rearing site as the most likely point of introduction of *S*. Typhimurium into the production system and subsequent dissemination to broiler flocks via the hatchery (28). However, phage typing and multiple-locus variable-number tandem-repeats analysis profiling of *S*. Typhimurium were used to determine the relationships between strains. Similar to the findings of this study, the circulation and spreading of identical *Salmonella* clones were reported throughout the broiler food chain and layer flocks (33). Moreover, breeders were evident to have a significant role in the vertical transmission and persistence of *Salmonella* within an integrated broiler production system and predominant identified serotypes were reported to be *S*. Typhimurium, *S*. Montevideo, *S*. Kentucky, and *S*. Enteritidis (27). However, the similarity of genomic profiles was determined based on the traditional PFGE approach, not with SNP-based analysis. In addition, the transport cages, and trailers can also serve as reservoirs for *Salmonella* and can disseminate pathogens to following broiler flocks (46).

Since the bacterium is well-known for both horizontal and vertical nature of transmission, it is possible that *Salmonella* can enter broiler complexes via multiple entry points along farms/facilities and finally accumulate in the processing plant and from there potentially contaminate raw poultry product. This may explain why we had the highest contamination rate in the post-pick carcass rinses as compared to other sample types. It is important to consider that once the bacterium contaminates facilities like

hatcheries, transport trucks, and processing plants, it can survive in those environments via biofilm formation for longer time periods (14) and can subsequently contaminate the product and can cause foodborne illness in humans. The different clusters of *S*. Kentucky strains and their close genetic relatedness in this study strongly indicate the possibility of transmission of same *Salmonella* strains between different stages, thus contaminating the final raw product in the processing plant. However, it is important to consider that this study was performed in a single commercial conventional broiler complex. The number of samples collected were representative samples from farms and facilities which had the mean parent flocks' (pullets and breeders) density of 12,200 birds per house and the mean broiler flocks' density of 33,750 birds per house. However, we sampled over 10% of each type of production farm (parent pullets, parent breeders, and broilers) and had three follow-up sampling visits from broiler farms to transport to processing plant. Although we collected four isolated colonies from one XLT4 plate of single culture positive samples, there is the possibility of missing minority serotypes over dominant serotypes on the petri plates since multi-serotypes were detected in a single sample (37, 47). Moreover, none of the samples were enumerated to estimate the *Salmonella* loads in each positive sample. In addition, out of 192 PCR-confirmed samples, only 100 samples were sent for WGS, and phylogenetic analyses were performed based on these 100 representative samples. For the samples sent off, we considered all 56 samples from production farms and hatcheries. while we considered only 44 selective representative samples from the transport and processing plant. Future studies may be directed to determine the transmission rates of *Salmonella* strains along the different stages of an integrated broiler complex and compare their virulence factors in relation to their transmission rates and persistence mechanisms.

## Conclusion

From the above results, the surroundings of facilities like hatcheries, transport, and processing plants are more likely to be contaminated with *Salmonella* as compared to parent flocks' and broiler grow-out flocks' surroundings. The contamination of facilities suggests deficiencies in the biosecurity measures along with cleaning and disinfection in such facilities. Since the microorganism can survive and persist in dry environmental conditions for longer time durations, it is possible that these facilities' surroundings can potentially serve as reservoirs for *Salmonella* to be disseminated to subsequent stages of an integrated broiler complex. Similarly, this study suggests multiple critical entry points and complex diversity of *Salmonella* serotypes along the different stages of the broiler production chain because the *Salmonella* serotypes present in the latter stages were not present in the upstream stages. Alternatively, most of the serotypes present in upstream stages were not found in the transport and processing plant.

In addition, the phylogenetic results suggest the possibility of transmission of the same *Salmonella* strains between outside and inside surroundings of farms/facilities and, between the different stages of the poultry production chain. From this study, it can be concluded that it is more likely that no single control strategy at a specific location will significantly minimize or eliminate *Salmonella* along the broiler production chain. However, the pathogen detection was significantly reduced (by 68%) in post-chill whole carcass rinses as compared to post-pick whole carcass rinses. Strict biosecurity measures, competitive exclusion with the use of pre/probiotics, acidified water during feed withdrawal, and vaccination of breeder hens are some common control strategies to minimize or eliminate this pathogen (48–50). Therefore, it is important to consider a comprehensive control strategy against *Salmonella* focusing on both facilities and production farms. Interestingly, the serotypes of foodborne importance such as *S*. Enteritidis, *S*. Typhimurium and *S*. Newport were more common in breeder farms and only *S*. Enteritidis was present in hatchery, broiler, and processing plant in this study. It would also be interesting to track chicks from hatchery to different broiler farms and compare the effects of varying management practices and how they influence *Salmonella* contamination. Furthermore, the inclusion of two more stages including further

processing and retail markets in this study may complete the whole poultry production chain from pullet farms to consumers. However, this study provides insights into the current *Salmonella* prevalence status among different stages of conventional broiler complex, possible entry points, and genetic relatedness among these strains.

## MATERIALS AND METHODS

### Study design

This epidemiological study was conducted from February to November 2023 in the South-East region of the United States. In total, an integrated broiler complex consisted of approximately 8 pullet farms (39 houses), 18 breeder farms (72 houses), 82 broiler farms (328 houses), 1 hatchery and 1 processing plant. However, in this study, altogether 1071 environmental samples were collected from outside and inside of 38 production houses (8 pullet, 10 breeder, and 20 broiler), a hatchery, 6 transport trucks, and a processing plant to isolate *Salmonella* (Table 7). Out of 8 pullet farms, 18 breeder farms and 82 broiler farms, we collected environmental samples from 4 pullet farms, 5 breeder farms, and 10 broiler farms with an aim to sample over 10% of each type of production farm.

### Production farm samples

Typically, all production farms consisted of two to six different houses within each farm. From which, two houses (furthest apart) from each farm were sampled. In total, 4 pullet farms at 2, 7, 10, 12, and 14 weeks of age, 5 breeder farms at 31, 45, 46, 52, 55, and 59 weeks of age, and 10 broiler farms at 14, 15, 16, 19, 21, 21, 26, 33, 35, and 47 days of age were sampled for this study (Tables 1 and 2). A boot swab, two sponge-stick swabs, two flypapers, two beetle traps, one litter grab, and one feed sample were collected from the inside environment of each poultry house. The researcher walked along the feeder and drinker lines from one end to other end of house wearing sterile booties/boot swabs (Envirobootie, Hardy Diagnostics, CA, USA) over plastic boot covers with an emphasis to step on fresh feces (21). The sterile 3M Sponge-Stick premoistened with 10 mL buffered peptone water broth (3M Food Safety, St. Paul, MN, USA) were swabbed against fan exhausts and house walls of area approximately 1,248 cm$^2$ (area of two A4 size papers; each with 21 cm × 29.7 cm) inside the chicken houses to capture dust

**TABLE 7** Number of environmental samples collected from farms and facilities in this study

| | Sample types | Samples for *Salmonella* isolation | | | | | | |
|---|---|---|---|---|---|---|---|---|
| | | Pullet | Breeder | Hatchery | Broiler | Transport | Processing | Total |
| 1 | Boot swabs | 8 | 10 | 8 | 23 | n/a[b] | 2 | 51 |
| 2 | Sponge-stick swabs | 18 | 39 | 28 | 38 | 58 | 43 | 224 |
| 3 | Beetle traps | 27 | 18 | 1 | 28 | n/a | n/a | 74 |
| 4 | Fly papers | 46 | 34 | 8 | 77 | n/a | n/a | 165 |
| 5 | Litter grab | 8 | 10 | n/a | 23 | n/a | n/a | 41 |
| 6 | Feed | 6 | 9 | n/a | 22 | n/a | n/a | 37 |
| 7 | Soil samples | 9 | 10 | 2 | 19 | n/a | n/a | 40 |
| 8 | Water puddles/drainage | 34 | 29 | 5 | 28 | n/a | n/a | 96 |
| 9 | Fecal sample | n/a | n/a | n/a | n/a | 35 | 13 | 48 |
| 10 | Carcass rinses (post-pick) | n/a | n/a | n/a | n/a | n/a | 64 | 64 |
| 11 | Carcass rinses (post-chill) | n/a | n/a | n/a | n/a | n/a | 96 | 96 |
| 12 | Parts rinses (post-treatment) | n/a | n/a | n/a | n/a | n/a | 20 | 20 |
| 13 | Miscellaneous[a] | 21 | 39 | 11 | 44 | n/a | n/a | 115 |
| | Total | 177 | 198 | 63 | 302 | 93 | 238 | 1,071 |

[a]Includes unknown feces, rodents' feces/intestines, ants, unknown feces, cattle feces, feather fluffs, wild bird feathers, and others that could potentially transmit *Salmonella* into the houses and facilities.
[b]Not available.

particles. Additionally, two sponge-stick swabs were also collected by swabbing the egg collection conveyor belt from each breeder house.

A week prior to the actual sampling day, four dual-sided fly papers (15 × 20 cm², Gideal, USA) and four beetle traps (two of each outside and inside each poultry house near the entrance) were placed with an objective to collect flies, dust particles and beetles or crawling insects. A polyvinyl chloridepipe of 23 cm long and 3.8 cm diameter with a rolled piece of cardboard (20 cm × 30 cm) inserted inside the pipe was used as a beetle trap (51). Similarly, the outside environmental samples of each house consisted of one soil sample near the entrance (10 g), water puddles/drainage, and miscellaneous samples along with fly papers and beetle traps. Fecal samples from domestic animals, wild birds, or unknown origin, dead rodents' intestines, ants, spiders, and wild bird feathers were categorized as miscellaneous sample types.

## Hatchery samples

The samples were collected from a multi-stage hatchery. As in the production farms, fly papers and beetle traps were sampled from the outside and inside environment. The major sample types from hatchery inside environment were sponge-stick swabs and boot swabs from various compartments like incubators, hatchers, egg buggies, chick tray washroom, chick separator room, along the passage between hatchers and incubators, and Embrex machine areas. Moreover, feather fluff from hatchers, composite broken eggs near Embrex machine area, and miscellaneous samples were also collected.

## Transport and processing of plant samples

There was a total of 12 different sampling visits. Among these, on last the three visits (10th, 11th, and 12th), the broiler flocks were followed to transport cage trucks and finally to the processing plant. As previously described, boot swabs, fly paper, and beetle traps were collected from broiler farms. Then, approximately 20 sponge-stick swab samples and 10 composite fecal samples were collected from transport. The sponge-stick swabs were swabbed on the floor of transport cages (approximately the area of two A4 size papers) just prior to loading of birds while fecal samples were collected from floor of transport cages and trailers.

From the processing plant, approximately 10 sponge stick swab samples and 5 fecal samples were collected from moving conveyor belt at live birds hang area in the processing plant per each visit. The swabbing area was approximately the area of two sheets of A4 size papers. In addition, 64 carcass rinses were collected after scalding and picking (after hock removal), and 96 post-chill carcass rinses were collected from three different sampling visits. For the collection of carcass rinses, 400 mL of sterilized buffered peptone water was poured into the rinse bag containing chicken carcass. It was shaken vigorously for 1 min and the rinsate was poured back into 400 mL containers and finally, placed on ice inside the cooler.

## Pathogen isolation

First, samples were pre-enriched with buffered peptone water (BPW) (Difco, Becton, Dickinson and Company, Sparks, MD, USA) with approximate dilution of 1:10 for all environmental samples except boot swabs, sponge-stick swabs, and carcass rinses. Boot swabs were pre-enriched with 100 mL BPW per sample, while sponge-stick swabs were pre-enriched with 50 mL BPW per sample. For carcass rinses as mentioned earlier, 400 mL of BPW rinsate was poured back into container. All the pre-enrichments were incubated for 18–24 h at 37°C and were screened with 3M MDS that works under the principle of loop-mediated isothermal conditions (52). USDA-FSIS MLG 4.13 method was used for the isolation of bacteria from MDS suspect positive samples (53). Briefly, those MDS-positive enrichments were subjected to selective enrichments: tetrathionate broth (TT) and Rappaport-Vassiliadis (RV) broth for 24 h at 41°C. The enrichments were then streaked on Xylose Lysine Tergitol agar 4 (XLT-4 Agar Base, Hardy Diagnostics, Santa Maria, CA,

USA) and then incubated for 24 h at 37°C to obtain the pure isolates. After that, four typical colonies from XLT4 were further re-streaked onto quadrants of XLT4 and Chrome Agar plates. The biochemical confirmation was performed with Triple Sugar Iron (TSI), Lysine Iron Agar (LIA), and Urease agar slants. Further, the black colonies, indicating hydrogen sulfide production, were subjected to an agglutination test with antiserum (DIFCO *Salmonella* O Antiserum Poly A—I and Vi; BD, Sparks, MD, USA) for *Salmonella* confirmation. Later, all pure isolates were stored in cryotubes containing Brucella broth (as non-selective enrichment media) with 20% glycerol at −80°C.

## DNA extraction, PCR confirmation, and sequencing

DNA was extracted using E. Z. N. A. Bacterial DNA Kit (Omega Bio-tek, Inc., Georgia, USA) from all 192 culture-positive samples. The DNA concentrations and quality parameters were measured using NanoDrop (Thermo Scientific Nanodrop One Spectrophotometer, MA, USA). The 260/280 ratio was between 1.8 and 2.0, while the 260/230 ratio was between 2.0 and 2.2. The PCR assay was performed for *invA* gene primers (Forward: GT GAAATTATCGCCACGTTCGGGCAA and Reverse: TCATCGCACCGTCAAAGGAACC). The PCR reaction mixture (10 µL) consisted of 5 µL GoTaq green master mix (Promega, Madison, WI, USA), 0.5 µL of 10 µM forward primer, 0.5 µL of 10 µM reverse primer, 3 µL of nuclease-free water, and 1 µL of template DNA. The reaction mixtures were subjected to Eppendorf master cycler (Eppendorf, Westbury, NY, USA) thermocycler under the conditions: 2 min at 95°C, 35 cycles of 30 s denaturation at 95°C, 30 s annealing at 55°C, and 90 s extension at 72°C, and final extension at 72°C for 5 min. The PCR products were subjected to electrophoresis on 2% agarose gels stained with SYBR safe DNA gel stain (Invitrogen, Carlsbad, CA, USA) and the gel was examined and photographed with (Bio-Rad ChemiDoc MP, Hercules, CA, USA). Out of 192 DNA samples, only 100 representative DNA samples were sent to SEQCENTER (Pittsburgh, PA, USA) for Illumina WGS with 100× coverage. For the selection of 100 representative samples, all 56 isolates were considered from production farms and a hatchery. However, the remaining 44 isolates were selected from transport and processing plant based on sample types and date of sample collection. At least three isolates were selected from sample types such as post-pick carcass rinses, swab from moving belt, and transport cages which had multiple isolates within the same sample type and date (as presented in Table S1).

## Bioinformatics analyses

Following WGS, FastQC (54) was used for quality checks of raw paired reads, while adapter trimming and quality trimming were performed using BBDuk tools (55). The *de novo* assembly was conducted using SPAdes (56) with a custom k-mer values of 25, 33, 55, 77, 95, and 127 kmer lengths. Quast with galaxy web interface was used in order to determine genome assembly statistics of each assembled genome (57). Moreover, the *Salmonella* serotypes were identified using SeqSero 1.2 service (58) of Center for Genomic Epidemiology (CGE) website. The raw reads and assembled genomes are available at NCBI GenBank under the BioProject accession number PRJNA1175719 .

## Genomic analyses of *Salmonella* strains

In addition to 100 *Salmonella* genomes from this study, 2,764 genomes of *Salmonella enterica* downloaded from NCBI were considered for building phylogenetic trees. Briefly, KmerFinder 3.2 service was used to find the closely related strain of specific serotypes, and SNP clusters were identified via SNP Tree Viewer for each serotype as mentioned in Table 8. A total of 1,651 *S*. Kentucky, 33 *S*. Enteritidis, 314 *S*. Alachua, 1,032 *S*. Typhimurium, 2 *S*. Mbandaka, 2 *S*. Montevideo, 17 *S*. Newport, 12 *S*. Senftenberg, 10 *S*. Uganda, 2 *S*. Ohio, and 3 *S*. Inverness assembled genomes were downloaded from NCBI for further phylogenetic analyses while in case of serotype Johannesburg SNP Tree Viewer was not available and SNP clusters could not be determined from NCBI.

TABLE 8  List of closest NCBI assemblies of *Salmonella* strains from this study for each serotype based on KmerFinder results and their respective SNP clusters on specific nodes in the SNP tree viewer

| Serotypes | KmerFinder results (assembly) | SNP clusters (specific node considered) |
|---|---|---|
| Ohio | GCF_003325335.1 | PDG000000002.3045/ PDS000037599.1 |
| Enteritidis | GCF_000624395.2 | PDG000000002.3045/ PDS000030237.1325 |
| | GCF_000623195.2 | PDG000000002.3045/ PDS000083226.661 |
| | GCF_000626275.2 | PDG000000002.3045/ PDS000032668.1187 |
| Kentucky | GCF_016454165.1 | PDG000000002.3045/ PDS000117429.153 |
| | GCF_001448485.2 | PDG000000002.3045/ PDS000032535.11 |
| Mbandaka | GCF_003641305.1 | PDG000000002.3045/ PDS000037663.1 |
| Inverness | GCF_001834735.2 | PDG000000002.3045/ PDS000007422.43 |
| Newport | GCF_000271905.2 | PDG000000002.3045/ PDS000127718.210 |
| Senftenberg | GCF_016454665.1 | PDG000000002.3045/ PDS000031981.14 |
| Typhimurium | GCF_005885875.1 | PDG000000002.3045/ PDS000026617.539 |
| Uganda | GCF_002507875.2 | PDG000000002.3045/ PDS000001852.762 |
| Alachua | GCF_016454225.1 | All available genomes |
| Montevideo | GCF_002761455.1 | PDG000000002.3045/ PDS000037243.1 |
| Johannesburg | GCF_001975405.1 | No SNP tree viewer |
| N/A or 9,46:z29:e,n,z15 | GCF_025311435.1 | No SNP tree viewer |

The SNP-based core genome phylogeny of all *Salmonella* strains was performed using ParSNP (59) and the phylogenetic trees were built using iTOL (60). *Salmonella enterica* LT2 strain from NCBI was used as the reference genome for building phylogeny. The SNP distance matrices among strains within each serotype was determined using CSI Phylogeny 1.4 service of CGE (61). The genetic relatedness among bacterial strains from this study were analyzed based on cutoff mutation of four SNPs within 30 days and nine SNPs for 120 days' time intervals to be considered as same strains (29).

## Statistical analyses

Data were analyzed using R version 4.3.1 (62) with generalized linear modeling for binomial distribution with logit link function where "1" and "0" were considered as the presence and absence of *Salmonella* in each sample respectively. The odds ratios and 95% confidence intervals were calculated for different stages (pullet, breeder, hatchery, broiler, transport, and processing plant) of poultry production, sample types (boot swabs, sponge-stick swabs, litter grabs, fly papers, beetle traps, soil, water puddles, feed, feces, post-pick and post-chill carcass rinses, and miscellaneous), environments (inside and outside), and seasons (spring, summer, and fall) using multivariable model. The samples from pullet, breeder, and broiler farms were grouped into production farms. The model assessed the effects of stages, sample types, environments, and seasons on the presence/absence of *Salmonella*. The level of significance ($a$) was set to 0.05. The pairwise comparison of variables within a group was estimated by the Tukey method to

separate means among different variables using "emmeans" package. The R codes used and corresponding results from the analysis are presented in supplementary materials.

## ACKNOWLEDGMENTS

The authors would like to acknowledge USDA-ARS for providing the fund, Department of Poultry Science, Auburn University, company personnel, and contract growers for their help during the study.

This work was supported by the United States Department of Agriculture Agricultural Research Service, Athens, GA (project numbers: 6040-32000-085-002-S and 6040-32000-012-006-S), the Alabama Agricultural Experiment Station, and the Hatch Program of the National Institute of Food and Agriculture, U.S. Department of Agriculture.

## AUTHOR AFFILIATIONS

[1]Department of Poultry Science, Auburn University, Auburn, Alabama, USA
[2]Deparment of Pathobiology, Auburn University, Auburn, Alabama, USA
[3]USDA ARS Poultry Microbiological Safety and Processing Research Unit, Athens, Georgia, USA
[4]Department of Poultry Science, Mississippi State University, Starkville, Mississippi, USA

## AUTHOR ORCIDs

Laura Huber  http://orcid.org/0000-0002-5138-0336
Kenneth S. Macklin  http://orcid.org/0000-0002-2707-8866

## FUNDING

| Funder | Grant(s) | Author(s) |
| --- | --- | --- |
| U.S. Department of Agriculture (USDA) | 6040-32000-085-002-S | Kenneth S. Macklin |
| U.S. Department of Agriculture (USDA) | 6040-32000-012-006-S | Dianna V. Bourassa |

## AUTHOR CONTRIBUTIONS

Yagya Adhikari, Data curation, Formal analysis, Investigation, Methodology, Project administration, Writing – original draft | Steven Kitchens, Investigation, Methodology, Writing – review and editing | Pankaj Gaonkar, Investigation, Methodology | Luis R. Munoz, Investigation, Methodology, Writing – review and editing | Stuart B. Price, Resources, Writing – review and editing | Dianna V. Bourassa, Funding acquisition, Resources | Laura Huber, Resources, Writing – review and editing | Richard J. Buhr, Funding acquisition, Resources, Writing – review and editing | Kenneth S. Macklin, Conceptualization, Funding acquisition, Supervision, Writing – review and editing.

## ADDITIONAL FILES

The following material is available online.

### Supplemental Material

**Supplemental figure and tables (Spectrum02090-24-s0001.docx).** Fig. S1 (A-K); Tables S1 and S2.

### Open Peer Review

**PEER REVIEW HISTORY (review-history.pdf).** An accounting of the reviewer comments and feedback.

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
