## [Reviewer comments · Microbiology Spectrum]

Microbiology Spectrum

Whole Genome Sequencing and Phylogenetic Analysis of *Salmonella* spp. Isolated from Pullets Through Final Raw Product in the Processing Plant of a Conventional Broiler Complex: A Longitudinal Study

Yagya Adhikari, Matthew Bailey, Steven Kitchens, Pankaj Gaonkar, Luis Munoz, Stuart Price, Diana Bourassa, Laura Huber, R Buhr, and Kenneth Macklin

Corresponding Author(s): Kenneth Macklin, Mississippi State University

Review Timeline:

Submission Date:	August 19, 2024
Editorial Decision:	October 15, 2024
Revision Received:	November 26, 2024
Accepted:	November 27, 2024

Editor: Artem Rogovsky

Reviewer(s): Disclosure of reviewer identity is with reference to reviewer comments included in decision letter(s). The following individuals involved in review of your submission have agreed to reveal their identity: Nurul Hawa Ahmad (Reviewer #1); Ben Davies Tall (Reviewer #2)

Transaction Report:

DOI: <https://doi.org/10.1128/spectrum.02090-24>

Re: Spectrum02090-24 (Whole Genome Sequencing and Phylogenetic Analysis of Salmonella spp. Isolated from Pullets Through Final Raw Product in the Processing Plant of a Conventional Broiler Complex: A Longitudinal Study)

Dear Prof. Kenneth Steven Macklin:

Thank you for the privilege of reviewing your work. Below you will find my comments, instructions from the Spectrum editorial office, and the reviewer comments.

Revision Guidelines

Sincerely,
Artem Rogovsky
Editor
Microbiology Spectrum

Reviewer #1 (Comments for the Author):

Authors have quantified the prevalence of Salmonella spp. in an integrated broiler complex and performed serotypes identification for isolates found at different stages of poultry production (from farms to processing plant) and related environmental samples. Authors further assessed the genetic relatedness of serotypes to track how Salmonella spp. contaminated the final products and recommended several directions for future studies.

Authors found that S. Kentucky was a predominant serotype but also emphasized that mixture of serotypes at the beginning stage of poultry production did not end up at the later stage. Based on SNP analysis, authors determined that the transmission of Salmonella spp. was coming from houses and facilities to final products.

Major comments:

1. L308-310: This statement, "More than 10% of total farms and facilities....", is a bit unclear. Did authors refer to 8 out of 39 pullet; 10 out of 72 breeder; 20 out of 328 broiler production houses?
2. L430: Justify the use of Brucella broth for storing pure isolates.
3. L483-485: Did authors mean logistic regression for data analysis?
4. L244-247: Given that Salmonella colonies were not labeled, it was hard to evaluate the masking effect of S. Kentucky strain on the media. It is suggested to include more justification related to physiological characteristics, stress response, adaptability, biofilm resistance of S. Kentucky in poultry environment as this strain was predominantly found.
5. Reference list should be formatted according to Spectrum requirements.

Minor comments:

1. L44: specify 'its' in this sentence for clarity
2. L84, L326: microorganism is more suitable than organism
3. L124-129: "95% CLs" can be omitted from the results description for simplicity as Table 4A contains detailed information.
4. L168: in or as?
5. L240-242: Remove these repetitive statements because the same idea has been mentioned in the results.
6. L295: remove "and" before "facilitate"
7. L318-319: "...and avoided only the repetitive samples from each samples type collected from transport and processing plant...." This part is confusing. What do authors mean by repetitive samples from each samples type?
8. L324: Specify "It" for clarity.
9. L367-371: Table 3A and 3B should be cited at the end of corresponding sentence for clarity.
10. Table 1: "Number of samples....." should be used. Salmonella spp. (spp. should not be italicized throughout the manuscript). Footnotes - instead of brackets, it is suggested to use symbol (such as *) to explain miscellaneous samples.
11. Table 4A - Footnotes description can be improved. "From the above table, the odds of Salmonella spp detection in hatchery, transport and processing plant samples were more likely ($P < 0.05$) as compared to environmental samples from production farms. Similarly, within the production farms, the odds of Salmonella detection in breeder farms' samples were more likely ($P < 0.05$) as compared to pullet and broiler farms' samples. However, there were no significant differences in the odds of Salmonella detection among different sample types, environments and seasons within the production farms." - these statements can be summarized and included in the results description. Use symbols, instead of brackets as mentioned in comment #10. Bold numbers should be indicated in footnotes.
12. Similar comment as #10 for Table 4B.
13. The P-value in the last column of Table 4A, in footnotes and main text should be consistently written.
14. Salmonella should not be underlined throughout the manuscript. In Figure 1, Salmonella should be italicized.

Reviewer #2 (Public repository details (Required)):

Genome sequences, bioproject and biosample numbers are needed.

Reviewer #2 (Comments for the Author):

Dear Editor:

I am sharing with you a review of Spectrum02090-24 manuscript entitled "Whole Genome Sequencing and Phylogenetic Analysis of Salmonella spp. Isolated from Pullets Through Final Raw Product in the Processing Plant of a Conventional Broiler Complex: A Longitudinal Study" which was submitted for publication in Microbiology Spectrum. The manuscript describes a study that determined the prevalence, critical entry points, and movement patterns of Salmonella enterica subspecies enterica serovars along different stages of a poultry production facility complex. The authors used the 3M Molecular Detection System to screen 1071 environmental samples which were collected from 38 production houses (eight pullet, ten breeder, and 20 broiler), a hatchery, six transport trucks, and a processing plant. Whole genome sequencing and phylogenetic analysis were performed to determine genetic relatedness among bacterial strains. The authors also used multivariable analysis to determine the odds ratios, and 95% confidence limits of comparing Salmonella detection for stages, 26 sample types, environments, and seasons ($\alpha < 0.05$). The odds of Salmonella detection were more likely ($p \leq 0.001$) in facilities like the hatchery, during transport, and within the processing plant as compared to detecting Salmonella in production farms such as pullet, breeder, and broiler farms. S. enterica serovar Kentucky was the most identified serovar found among 12 different serovars. Phylogenetic analysis showed a strong genetic relationship among bacterial strains isolated from the various stages which also suggested diverse movement patterns among the bacterial strains and the possibility of multiple critical points which led to bacterial entry into the complex. The goal of my review is to offer suggestions to the authors to make the manuscript ready for publication. However, at this time, I cannot advocate for publishing this manuscript in its current form. These comments are given to the authors so that they may augment their already valuable content. I am listing my comments according to line number. The following are my comments and suggestions.

- 1) Abstract line 17, I suggest that the authors revise the sentence to read as: Salmonella are Gram-negative, rod-shaped, entero-invasive foodborne bacteria and are frequently detected in chicken houses and facilities of poultry broiler complexes.
- 2) Line 19, Are the authors referring to Salmonella enterica, Salmonella bongori, and the newly recognized species, Salmonella subterranean, which was recognized in 2005. The use of the abbreviation, spp., here and throughout the manuscript is misleading because the results reported were only for serovars of Salmonella enterica subspecies enterica. I suggest that the authors delete the spp. abbreviation.
- 3) Line 22-23, The use of (MDS) and (WGS) are not needed because they are not referenced further in the abstract.
- 4) Line 33, Do the authors mean patterns?
- 5) Line 34, Do the authors mean ...points for bacterial pathogens entering the complex?
- 6) Line 36, Do the authors meancontaminate the final....
- 7) Line 44, Do the authors mean patterns instead of pattern?
- 8) Line 53, I suggest that the authors revise to read as: ... multiple critical points for the entry of pathogens ...
- 9) Line 63, refer to the previous comment regarding spp.
- 10) Line 69, I think the authors meant to write: Salmonella are Gram-negative, facultatively anaerobic, non-spore-forming, rod-shaped bacteria of the Enterobacteriaceae family.
- 11) Line 105, What about the persistence of strains and serovars within the complex?
- 12) Line 107, See earlier comment in comment 4.
- 13) Line 108-109, This is repetitive with line 107. I would combine sentences. Maybe something like: The objective of this longitudinal study was to determine the prevalence, critical entry points, phylogenetic relatedness, and movement patterns of Salmonella along the different stages of an integrated broiler production complex to improve pathogen control strategies.
- 14) Lines 114, 116, and 119, I suggest replacing "on" with "by"
- 15) Line 162, This is an assumption. I suggest just writing that the serotypes were not found in transport and processing plants.
- 16) Line 244, I suggest that the authors delete serotype here.
- 17) Line 247, Metagenomic Sequence analysis may be helpful here. I am not suggesting that the authors perform such for this study but it could be mentioned here for future studies.
- 18) Line 332, I suggest that the authors revise this sentence to read as.... most of the serotypes present in upstream stages were not found in the transport and processing plant's sampling stages.
- 19) Line 338, I suggest replacing food with poultry production to read as poultry production chain.
- 20) Line 343, I think here the authors could offer suggestions as to how to implement these steps and what other control steps could be described.
- 21) Line 346, Does this statement suggest that vaccination is not efficacious in reducing S. Enteritidis?
- 22) Line 385, what was the size of the soil sample?
- 23) Line 416, I think the authors need to describe in more detail the dilution ratios of each sample type used for the enrichment?
- 24) Line 428, I think the authors meant to write agglutination instead of precipitation? Was the colonies subjected to growth on a non-selective agar before testing?
- 25) Line 473, There no mention of submission of genomes to NCBI. What were the Bioproject and biosamples numbers used for submission? This information is mandated by the journal.
- 26) Line 511, I think the authors meant to write funding instead of funds.
- 27) Figs. 4 and 5, It would be very informative if Sequence Type information were overlaid on these trees. These analyses would augment the suggested extensive genome-wide sequence differences among the strains that were not fully captured by conventional SNP techniques. This would also augment emergent properties within these sub-groups of spatially and temporally discrete potential lineages as they underwent robust, quantifiable micro-evolutionary changes. Added to this would be the detection of plasmids, AMR, and prophage genes. These analyses would also support movement patterns, persistence, and survival of the various strains throughout the complex.

Nov. 22, 2024

Dear Managing Editor

Below are our comments to the paper we submitted “Whole genome sequencing and phylogenetic analysis of *Salmonella* spp. isolated from pullets through final raw product in the processing plant of a conventional broiler complex: a longitudinal study” and we addressed all the reviewers comments and answered their questions. We appreciate their efforts. - KSM

Reviewer #1 (Comments for the Author):

Authors have quantified the prevalence of *Salmonella* spp. in an integrated broiler complex and performed serotypes identification for isolates found at different stages of poultry production (from farms to processing plant) and related environmental samples. Authors further assessed the genetic relatedness of serotypes to track how *Salmonella* spp. contaminated the final products and recommended several directions for future studies.

Authors found that S. Kentucky was a predominant serotype but also emphasized that mixture of serotypes at the beginning stage of poultry production did not end up at the later stage. Based on SNP analysis, authors determined that the transmission of *Salmonella* spp. was coming from houses and facilities to final products.

Major comments:

1. L308-310: This statement, "More than 10% of total farms and facilities....", is a bit unclear. Did authors refer to 8 out of 39 pullet; 10 out of 72 breeder; 20 out of 328 broiler production houses?

Corrected. Out of 8 pullet farms, 18 breeder farms and 82 broiler farms, we collected environmental samples from 4 pullet farms, 5 breeder farms and 10 broiler farms with an aim to sample over 10% of each type of production farm.

2. L430: Justify the use of Brucella broth for storing pure isolates.

Brucella broth is an enriched non-selective media often used to cultivate *Brucella* species and other fastidious microbes. Though *Salmonella* is not fastidious bacteria and can grow in wide range of enriched media, the purpose of using brucella broth in this study was just as enriched non-selective media.

3. L483-485: Did authors mean logistic regression for data analysis?

Yes. Data were analyzed with specific type of GLM for binomial outcomes with logit link function to predict the probability of the event occurring. However, GLM for binomial distribution allows for other link functions beyond the logit, making it more flexible.

4. L244-247: Given that *Salmonella* colonies were not labeled, it was hard to evaluate the masking effect of S. Kentucky strain on the media. It is suggested to include more justification

related to physiological characteristics, stress response, adaptability, biofilm resistance of S. Kentucky in poultry environment as this strain was predominantly found.

S. Kentucky is most frequently isolated serotype from poultry in the U.S. The reason may be due to its unique selective adaptational growth that could outcompete other serotypes because Kentucky strains were reported to persist longer and at higher cell densities in the ceca of birds as compared to Typhimurium strains (Cheng et al., 2015). Moreover, stationary sigma factor *rpoS*- regulated genes associated with stress response as well as metabolic responses were highly expressed in Kentucky strains as compared to Typhimurium strains. (Cheng et al, 2015). Similarly, Guillén et al. (2020) reported that Kentucky strains showed lower acid resistance and higher heat resistance, and similar oxidative resistance as compared to Enteritidis strains. It has also been shown that Kentucky strains might outcompete other serotypes present in a single sample during selective enrichment with RV or TT broth (Cox et al. 2019). Another possibility can be that the pre-enrichment promoted abundant growth of Kentucky serotype which masked the detection of other serotypes during conventional culture method. ----- was added in the discussion section

Interestingly, though the incidence of *Salmonella* was significantly reduced from post-pick to post-chill carcass rinses, further research may be required to determine how the 5 Kentucky isolates which were supposed to be relatively fragile serotype survived during chilling step with antimicrobials and recovered from post-chill carcass rinses. ----was added at the end of discussion section

5. Reference list should be formatted according to Spectrum requirements.

Corrected

Minor comments:

1. L44: specify 'its' in this sentence for clarity

Corrected

2. L84, L326: microorganism is more suitable than organism

Corrected

3. L124-129: "95% CLs" can be omitted from the results description for simplicity as Table 4A contains detailed information.

Corrected

4. L168: in or as?

Corrected

5. L240-242: Remove these repetitive statements because the same idea has been mentioned in the results.

Corrected

6. L295: remove "and" before "facilitate"

Corrected

7. L318-319: "...and avoided only the repetitive samples from each samples type collected from transport and processing plant...." This part is confusing. What do authors mean by repetitive samples from each samples type?

Corrected

8. L324: Specify "It" for clarity.

Corrected

9. L367-371: Table 3A and 3B should be cited at the end of corresponding sentence for clarity.

Corrected

10. Table 1: "Number of samples....." should be used. *Salmonella* spp. (spp. should not be italicized throughout the manuscript). Footnotes - instead of brackets, it is suggested to use symbol (such as *) to explain miscellaneous samples.

Corrected

11. Table 4A - Footnotes description can be improved. "From the above table, the odds of *Salmonella* spp detection in hatchery, transport and processing plant samples were more likely ($P < 0.05$) as compared to environmental samples from production farms. Similarly, within the production farms, the odds of *Salmonella* detection in breeder farms' samples were more likely ($P < 0.05$) as compared to pullet and broiler farms' samples. However, there were no significant differences in the odds of *Salmonella* detection among different sample types, environments and seasons within the production farms." - these statements can be summarized and included in the results description. Use symbols, instead of brackets as mentioned in comment #10. Bold numbers should be indicated in footnotes.

Corrected

12. Similar comment as #10 for Table 4B.

Corrected

13. The P-value in the last column of Table 4A, in footnotes and main text should be consistently written.

Corrected

14. *Salmonella* should not be underlined throughout the manuscript. In Figure 1, *Salmonella* should be italicized.

Corrected

Reviewer #2 (Public repository details (Required)):

Genome sequences, bioproject and biosample numbers are needed.

All the raw reads (fastq files) and assembled genomes (fasta files) were submitted under NCBI BioProject Accession no. PRJNA1175719.

Reviewer #2 (Comments for the Author):

Dear Editor:

I am sharing with you a review of Spectrum02090-24 manuscript entitled "Whole Genome Sequencing and Phylogenetic Analysis of Salmonella spp. Isolated from Pullets Through Final Raw Product in the Processing Plant of a Conventional Broiler Complex: A Longitudinal Study" which was submitted for publication in Microbiology Spectrum. The manuscript describes a study that determined the prevalence, critical entry points, and movement patterns of Salmonella enterica subspecies enterica serovars along different stages of a poultry production facility complex. The authors used the 3M Molecular Detection System to screen 1071 environmental samples which were collected from 38 production houses (eight pullet, ten breeder, and 20 broiler), a hatchery, six transport trucks, and a processing plant. whole genome sequencing and phylogenetic analysis were performed to determine genetic relatedness among bacterial strains, The authors also used multivariable analysis to determine the odds ratios, and 95% confidence limits of comparing Salmonella detection for stages, 26 sample types, environments, and seasons ($\alpha < 0.05$). The odds of Salmonella detection were more likely ($p \leq 0.001$) in facilities like the hatchery, during transport, and within the processing plant as compared to detecting Salmonella in production farms such as pullet, breeder, and broiler farms. S enterica serovar Kentucky was the most identified serovar found among 12 different serovars. Phylogenetic analysis showed a strong genetic relationship among bacterial strains isolated from the various stages which also suggested diverse movement patterns among the bacterial strains and the possibility of multiple critical points which led to bacterial entry into the complex. The goal of my review is to offer suggestions to the authors to make the manuscript ready for publication. However, at this time, I cannot advocate for publishing this manuscript in its current form. These comments are given to the authors so that they may augment their already valuable content. I am listing my comments according to line number. The following are my comments and suggestions.

1) Abstract line 17, I suggest that the authors revise the sentence to read as: Salmonella are Gram-negative, rod-shaped, entero-invasive foodborne bacteria and are frequently detected in chicken houses and facilities of poultry broiler complexes.

Corrected

2) Line 19, Are the authors referring to Salmonella enterica, Salmonella bongori, and the newly recognized species, Salmonella subterranean, which was recognized in 2005. The use of the abbreviation, spp., here and throughout the manuscript is misleading because the results reported were only for serovars of Salmonella enterica subspecies enterica. I suggest that the authors delete the spp. abbreviation.

Corrected

3) Line 22-23, The use of (MDS) and (WGS) are not needed because they are not referenced further in the abstract.

Corrected

4) Line 33, Do the authors mean patterns?

Yes. Corrected

5) Line 34, Do the authors mean ...points for bacterial pathogens entering the complex?

Yes. Corrected

6) Line 36, Do the authors meancontaminate the final....

Yes. Corrected

7) Line 44, Do the authors mean patterns instead of pattern?

Yes. Corrected

8) Line 53, I suggest that the authors revise to read as: ... multiple critical points for the entry of pathogens ...

Corrected

9) Line 63, refer to the previous comment regarding spp.

Corrected

10) Line 69, I think the authors meant to write: Salmonella are Gram-negative, facultatively anaerobic, non-spore-forming, rod-shaped bacteria of the Enterobacteriaceae family.

Corrected

11) Line 105, What about the persistence of strains and serovars within the complex?

Corrected

12) Line 107, See earlier comment in comment 4.

Corrected

13) Line 108-109, This is repetitive with line 107. I would combine sentences. Maybe something like: The objective of this longitudinal study was to determine the prevalence, critical entry points, phylogenetic relatedness, and movement patterns of Salmonella along the different stages of an integrated broiler production complex to improve pathogen control strategies.

Corrected

14) Lines 114, 116, and 119, I suggest replacing "on" with "by"

Corrected

15) Line 162, This is an assumption. I suggest just writing that the serotypes were not found in transport and processing plants.

Corrected

16) Line 244, I suggest that the authors delete serotype here.

Corrected

17) Line 247, Metagenomic Sequence analysis may be helpful here. I am not suggesting that the authors perform such for this study but it could be mentioned here for future studies.

Corrected

18) Line 332, I suggest that the authors revise this sentence to read as.... most of the serotypes present in upstream stages were not found in the transport and processing plant's sampling stages.

Corrected

19) Line 338, I suggest replacing food with poultry production to read as poultry production chain.

Corrected

20) Line 343, I think here the authors could offer suggestions as to how to implement these steps and what other control steps could be described.

The control steps were mentioned in the previous sentence.

21) Line 346, Does this statement suggest that vaccination is not efficacious in reducing S. Enteritidis?

No. We just wanted to convey the brief information about serotypes of foodborne importance identified in this study. However, live ST vaccine was provided at day 1 and inactivated SE vaccine was provided around week 10-12 in these birds.

22) Line 385, what was the size of the soil sample?

Corrected (10 grams)

23) Line 416, I think the authors need to describe in more detail the dilution ratios of each sample type used for the enrichment?

Corrected

24) Line 428, I think the authors meant to write agglutination instead of precipitation? Was the colonies subjected to growth on a non-selective agar before testing?

Corrected. Yes. We grew the isolates in tryptic soy broth and inoculated them in three different slants. Next, the left over broth was used for agglutinations.

25) Line 473, There no mention of submission of genomes to NCBI. What were the Bioproject and biosamples numbers used for submission? This information is mandated by the journal.

26) Line 511, I think the authors meant to write funding instead of funds.

We think fund better suits in the sentence.

27) Figs. 4 and 5, It would be very informative if Sequence Type information were overlaid on these trees. These analyses would augment the suggested extensive genome-wide sequence differences among the strains that were not fully captured by conventional SNP techniques. This would also augment emergent properties within these sub-groups of spatially and temporally discrete potential lineages as they underwent robust, quantifiable micro-evolutionary changes.

Added to this would be the detection of plasmids, AMR, and prophage genes. These analyses would also support movement patterns, persistence, and survival of the various strains throughout the complex.

All the 64 Kentucky strains belonged to ST-152. Therefore, it was not overlaid on the Kentucky phylogenetic tree. Similarly, all remaining serotypes also had common sequence type for each serotype except Enteritidis. Therefore, sequence types results were not included in the phylogenetic trees.

In addition, we plan to submit a separate manuscript for publication with genotypic and phenotypic AMR results and plasmid results. Therefore, we have not included those results in this manuscript.

Sequence Types	Serotypes	Salmonella Strains
ST152	Kentucky (64)	Remaining all
ST11	Enteritidis (7)	MACK3, MACK4, MACK13, MACK15, MACK42, MACK73, MACK96
ST11	Enteritidis (1)	MACK105
ST6	Enteritidis (1)	MACK6
ST1298	Alachua (6)	MACK58, MACK63, MACK75, MACK76, MACK77, MACK78
ST471	Johannesburg (3)	MACK1, MACK17, MACK18
ST19	Typhimurium (3)	MACK43, MACK44, MACK45
ST2404	Mbandaka (3)	MACK19, MACK21, MACK22
ST81	Montevideo (3)	MACK106, MACK107, MACK108
ST118	Newport (2)	MACK24, MACK25
ST684	Uganda (2)	MACK57, MACK89
ST1610	9:z29:- (2)	MACK10, MACK11
ST1384	Inverness (1)	MACK23
ST14	Senftenberg (1)	MACK31
	Total (100)	

Note: Figure 6 was added to the manuscript because we think that it would be helpful for the readers to visualize the movement patterns of *Salmonella* strains along different stages of an integrated broiler complex that we observed in this study.

Re: Spectrum02090-24R1 (Whole Genome Sequencing and Phylogenetic Analysis of Salmonella spp. Isolated from Pullets Through Final Raw Product in the Processing Plant of a Conventional Broiler Complex: A Longitudinal Study)

Dear Prof. Kenneth Steven Macklin:

Your manuscript has been accepted, and I am forwarding it to the ASM production staff for publication. Your paper will first be checked to make sure all elements meet the technical requirements. ASM staff will contact you if anything needs to be revised before copyediting and production can begin. Otherwise, you will be notified when your proofs are ready to be viewed.

Sincerely,
Artem Rogovsky
Editor
Microbiology Spectrum